# Asymmetric Fiber Supercapacitors Based on a FeC₂O₄/FeOOH-CNT Hybrid Material

Paa Kwasi Adusei [1], Kevin Johnson [2], Sathya N. Kanakaraj [1], Guangqi Zhang [1], Yanbo Fang [1], Yu-Yun Hsieh [1], Mahnoosh Khosravifar [1], Seyram Gbordzoe [1], Matthew Nichols [2] and Vesselin Shanov [1,2,*]

1   Department of Mechanical and Materials Engineering, University of Cincinnati, Cincinnati, OH 45221-0072, USA; aduseipi@mail.uc.edu (P.K.A.); kanakasa@mail.uc.edu (S.N.K.); zhangg5@mail.uc.edu (G.Z.); fangyb@mail.uc.edu (Y.F.); hsiehyu@mail.uc.edu (Y.-Y.H.); khosramh@mail.uc.edu (M.K.); gbordzsm@mail.uc.edu (S.G.)
2   Department of Chemical and Environmental Engineering, University of Cincinnati, Cincinnati, OH 45221-0012, USA; johns6kj@mail.uc.edu (K.J.); nichomp@mail.uc.edu (M.N.)
*   Correspondence: Vesselin.shanov@uc.edu; Tel.: +1-513-556-2461

**Abstract:** The development of new flexible and lightweight electronics has increased the demand for compatible energy storage devices to power them. Carbon nanotube (CNT) fibers have long been known for their ability to be assembled into yarns, offering their integration into electronic devices. They are hindered, however, by their low intrinsic energy storage properties. Herein, we report a novel composite yarn, synthesized through solvothermal processes, that attained energy densities in the range between 0.17 μWh/cm² and 3.06 μWh/cm², and power densities between 0.26 mW/cm² and 0.97 mW/cm², when assembled in a supercapacitor with a PVDF-EMIMBF4 electrolyte. The created unique composition of iron oxalate + iron hydroxide + CNT as an anode worked well in synergy with the much-studied PANI + CNT cathode, resulting in a highly stable yarn energy storage device that maintained 96.76% of its energy density after 4000 cycles. This device showed no observable change in performance under stress/bend tests which makes it a viable candidate for powering wearable electronics.

**Keywords:** CNT fibers; FeC₂O₄/FeOOH-CNT hybrid fiber; solvothermal processing; fiber supercapacitors





## 1. Introduction

The rapid development of wearable electronics and flexible energy harvesters has fueled great interest in fiber and yarn based energy storage devices [1–3]. Among them supercapacitors have been the subject of intense research since they combine the properties of high energy batteries and high-power dielectric capacitors into one unit.

Supercapacitors consist of two electrodes, an electrolyte, and a separator. The electrodes are generally classified as carbon-based materials or pseudocapacitive materials [4,5]. Carbon-based electrodes are widely used in electric double layer capacitors (EDLCs) due to their large specific surface area, high electrical conductivity, and low cost [6,7]. EDLCs attain high power densities and cycling stability. However, they have been proven to produce comparatively lower energy densities [8,9]. The lower energy density is ascribed to the charge-storage mechanism employed by EDLCs, which is based on electrosorption of charges at the electrode-electrolyte interface. Conversely, pseudocapacitive devices (typically made of transition metal oxides and conductive polymer electrodes) are able to attain larger energy densities achieved by storage of charges in the electrodes as a response to fast surface and near-surface redox reactions [6,10–12].

The major drawback of supercapacitors is their relatively low energy densities, and the increase of the latter has been the focus of growing research. One of the most promising

strategies to boost energy density is the fabrication of asymmetric and hybrid supercapacitors. These supercapacitors combine faradaic and capacitive electrodes, thereby aiming to increase the energy density while simultaneously maintaining high power densities by leveraging the different charge-storage mechanisms of the capacitive and faradaic electrodes [9,13]. The added advantage in such an arrangement is that these devices reveal an extended operating potential window (up to 2 V) in aqueous electrolytes [14,15].

In recent years, there has been an increase in the work done on fiber-based supercapacitors. However, one of the least explored areas is in the development of Anode based materials. This is due to several factors such as Anodic materials (including VN [16,17], $MoO_3$ [18–21], $WO_3$ [22,23], and $Bi_2O_3$ [24–26]) which are very difficult to synthesize and are extremely unstable. There are two main approaches to asymmetric supercapacitors widely used in the literature: (a) the use of Cathodic materials at the Anode [27] and (b) the use of carbon-based materials as anodes [28]. One of the largest disadvantages/limitations to using cathodic materials as anodes is that they are unstable at negative potentials in aqueous electrolytes (this limits the voltage window the device can operate at) [29–31].

Iron-based materials such as iron oxides ($Fe_2O_3$, $Fe_3O_4$), iron oxalate ($FeC_2O_4$), and iron oxide hydroxide (FeOOH) have emerged as good candidates for anode materials due to their wide operating voltage windows, natural abundance, low cost, environmental friendliness, and high theoretical capacitance [14,32,33]. There are, however, significant challenges in translating these desirable qualities into practical applications due to their relatively small surface area, low electrical conductivity (due to kinetic limitations), poor rate performance, and limited cyclic stability. The latter is originating from volume expansion during cycling of these materials, which along with their intrinsic properties adversely affects their pseudocapacitive performance [34–36].

In order to overcome these challenges, several strategies have been successfully implemented. First, the use of iron oxide and iron oxide hydroxide as nanoscale size components for anodes [37,38] has been found to yield great results. These nanomaterials enable the shortening of the diffusion distance, offer increased surface area for better electrode-electrolyte contact, and facilitate proton transfer [37]. Second, a combination of these materials with carbonaceous materials improves the overall electrochemical performance since the carbon-based materials can increase the electrical conductivity and contribute to the rate and cycling ability. The carbonaceous materials can also serve as a scaffold to host the nanoscale iron oxide and iron oxide hydroxide structures.

In this work, we report a $CNT/FeC_2O_4$ + FeOOH anode material with an excellent electrochemical performance (high specific capacitance of 324 $mF/cm^2$ at 2 $mA/cm^2$) at a wide potential voltage window for use in an asymmetric supercapacitor. A solvothermal set-up has been employed to grow a mixture of iron oxalate ($FeC_2O_4$) and iron oxide hydroxide (FeOOH) nanoparticles on oxygen plasma functionalized CNT (OPFCNT) fibers. As reported in the literature, low-crystalline metal oxides with increased structural defects and disorder have enabled energy storage devices to achieve better cycling stability [39,40].

The cathode material studied in our asymmetric supercapacitor was polyaniline (PANI) decorated CNT fibers. PANI is a polymer revealing high electronic conductivity, redox, and ion exchange properties, ease of preparation, and excellent environmental stability. All of these qualities have been extensively explored and employed in energy storage devices [41–43].

## 2. Experimental

### 2.1. Fabrication of OPFCNT Fiber

Spinnable CNT arrays have been grown on oxidized silicon wafers. A buffer layer of 5 nm $Al_2O_3$ was deposited on the surface of the silicon wafer by magnetron sputtering. Next, a layer of Fe and Co with a total thickness of 1.2 nm was sputtered on the $Al_2O_3$ film to serve as catalysts for the growth of vertically aligned CNT arrays in an ET3000 CVD reactor from CVD Equipment Corp. There, the silicone substrate was exposed to a mixture of methane, argon, and hydrogen at 750 °C, which yielded aligned double

and triple-wall CNTs with length of about 300 μm. Details of the growth process and properties of the array can be found in our previous work [44]. The resulting CNT array was spinnable and by using an inhouse set-up, we were able to spin fibers of varying lengths and diameters [45,46].

This CNT fiber was then functionalized by using atmospheric pressure oxygen plasma via gradually pulling it with a speed of 0.21 cm/s through a tubular plasma head (Surfx Atomflo 400 system) operated at 60 W power while flowing 0.1 L/min oxygen mixed with 15 L/min helium. Further details on the plasma functionalization process and set-up have been described in our previous work [47]. These process steps are schematically shown in Figure 1.

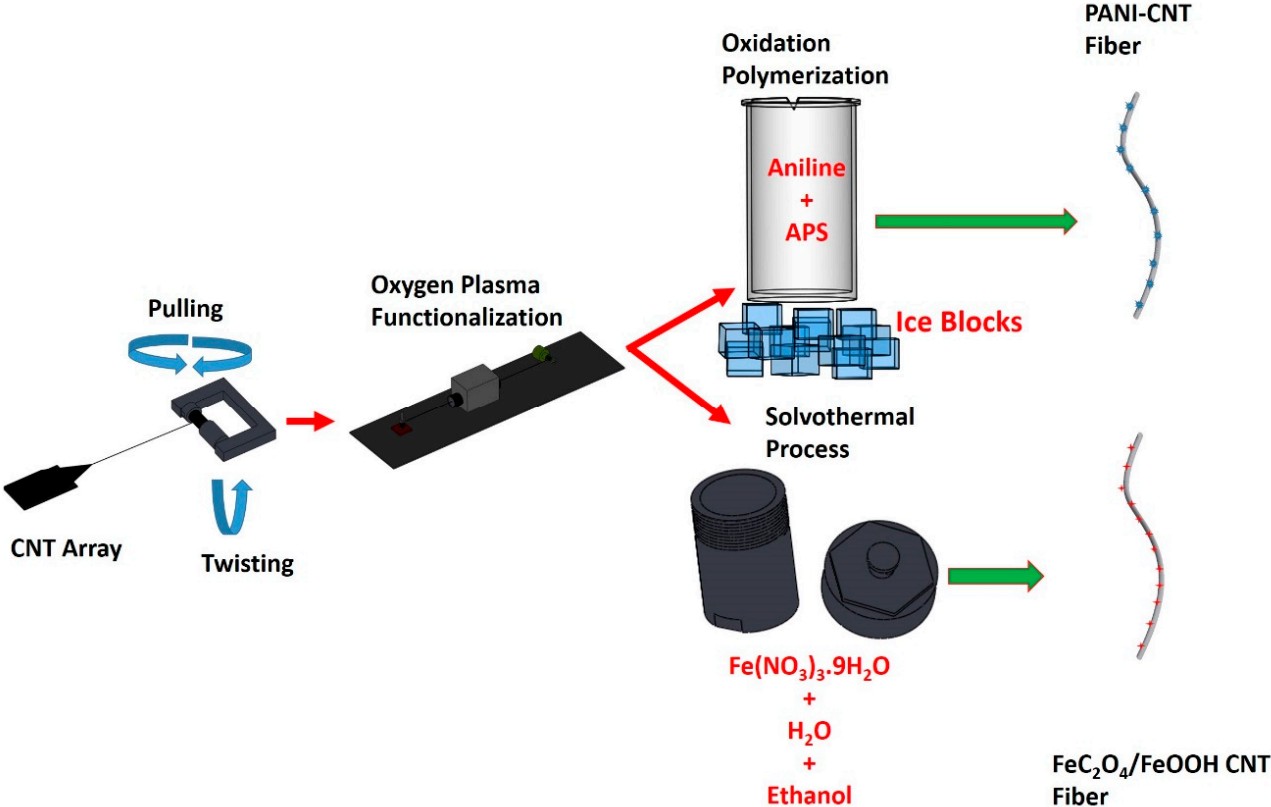

**Figure 1.** Fiber fabrication process steps.

### 2.2. *FeC$_2$O$_4$ + FeOOH Deposition on OPFCNT Fiber for Anode Formation*

Fe(NO$_3$)$_3$·9H$_2$O of varying amounts was dissolved in 30 mL of distilled water and 30 mL ethanol and then stirred for 2 h. The resultant clear solution was transferred into a Teflon-lined stainless-steel autoclave along with the OPFCNT fiber. The solvothermal process was carried out at 120 °C over 16 h. The fiber was then cooled down and rinsed with distilled water. Finally, it was dried in an oven at 70 °C for 12 h. The solvothermal process is also illustrated in Figure 1.

### 2.3. *PANI Deposition on CNT Fiber for Cathode Formation*

The PANI-CNT hybrid fiber was synthesized by placing portions of the OPFCNT in a beaker with 1 M aniline monomer dissolved in 1 mol/L HCl and 1 M ammonium peroxydisulphate (APS) in 1 mol/L HCl, as reported in our previous work [48]. The hybrid fiber was created at a 2:1 ratio (aniline monomer to APS). The duration during which the fibers were left in the solution was for 15 min and 60 min. These times were selected to control the mass loading of PANI on the fiber. This processing yielded PANI nanorods formed on the surface of the OPFCNT fiber, which were characterized by Raman

spectroscopy and SEM. The hybrid PANI-CNT fiber was then used as the cathode in the asymmetric supercapacitor.

### 2.4. Electrode Fabrication

All electrodes and devices were made using 7.5 cm long CNT fibers. These fibers were attached to copper tapes with fast drying silver paint (TedPella Inc., Redding, CA, USA). The copper tape served as an electrically conducting lead to the electrochemical testing instrument. In the fabricated device, the electrodes were coated with a gel electrolyte comprising of PVA-KOH. After curing, they were placed parallel to each other on a cellulose weighing paper (LabExact). The gel electrolyte was added to this configuration of fibers and sealed with a Kapton tape to create a full device (Supplementary Material, Figure S1).

### 2.5. Characterization of Supercapacitor Electrodes

Scanning electron microscopy (SEM) (FEI SCIOS dual beam, 5 kV) and surface enhanced Raman spectroscopy (SERS) using a custom Nikon Ti Eclipse inverted dark-field microscope with a Nikon darkfield condenser (Ni-E 0.95-0.8) [49], have been conducted. Raman spectroscopy (Renishaw inVia, equipped with 514 nm Ar-ion laser and a laser spot of ~1 $\mu m^2$) was also used to characterize the synthesized PANI-based electrodes.

X-ray photoelectron spectroscopy (XPS) (VG Thermo-Scientific, Waltham MA MultiLab 3000 ultra-high vacuum surface analysis system) was done using an Al K$\alpha$ source of 1486.6 eV excitation energy. The fibers were crushed in a hot press to create a mat-like material for X-ray diffraction in a Philips X'Pert Pro PW3040 PANalytical.

Electrochemical measurements were carried out for the fiber electrodes with a Gamry Interface 1000 electrochemical workstation at room temperature. The CNT fiber underwent cyclic voltammetry, galvanostatic charge-discharge, and electrochemical impedance spectroscopy measurements over a frequency range from $10^5$ to $10^{-1}$ Hz at a sinusoidal voltage amplitude of 10 mV and an open circuit potential. The capacitance (C) of the devices and of the electrodes were calculated using the equation C = I$\Delta$t/$\Delta$V. The areal capacitance (C$_A$) was calculated by the following formula: $C_A = C_{device}/A$. The areal energy density (E$_A$) and power density (P$_A$) were computed by the expressions: $E_A = \frac{1}{2} \times \frac{C_A(\Delta V)^2}{3.6}$ and $P_A = 3600\frac{E_A}{t}$ where I is the discharge current, t is the discharge time, $\Delta$V is the operating voltage window, A refers to the area of the device [11,14,48].

## 3. Results and Discussion

### 3.1. Negative Electrode (Anode)

SEM images of the OPFCNT and the CNT/FeC$_2$O$_4$ + FeOOH material are presented in Figure 2a–d. Figure 2a,c display the OPFCNT. The high magnification in Figure 2c reveals the nanotube bundles assembled within the fiber. In Figure 2b is shown the SEM images of the FeC$_2$O$_4$/FeOOH/CNT fiber. It can be noted a reduction in the diameter of this fiber compared to the one in Figure 2a. This is due to the densification the fiber underwent when interacting with a solvent-a process which leads to capillary force action on the tubes, causing the fiber to shrink in diameter [46,50]. Looking at high magnification SEM images (Figure 2d), we were unable to see pronounced individual carbon nanotubes. In the same Figure 2d, one can observe the nanostructured round particles of FeC$_2$O$_4$/FeOOH with nanometer size.

These particles were proven to be FeC$_2$O$_4$/FeOOH by the conducted X-ray diffraction study. The XRD pattern of the fiber is displayed in Figure 3a (PDF cards no: 23-0293, 29-0713, and 26-1080). Figure 3b presents the energy dispersive spectroscopy (EDS) data of the same fiber. The elemental percentage by weight of the detected elements was Fe-14.25%, C-67.27%, and oxygen-18.48%, respectively.

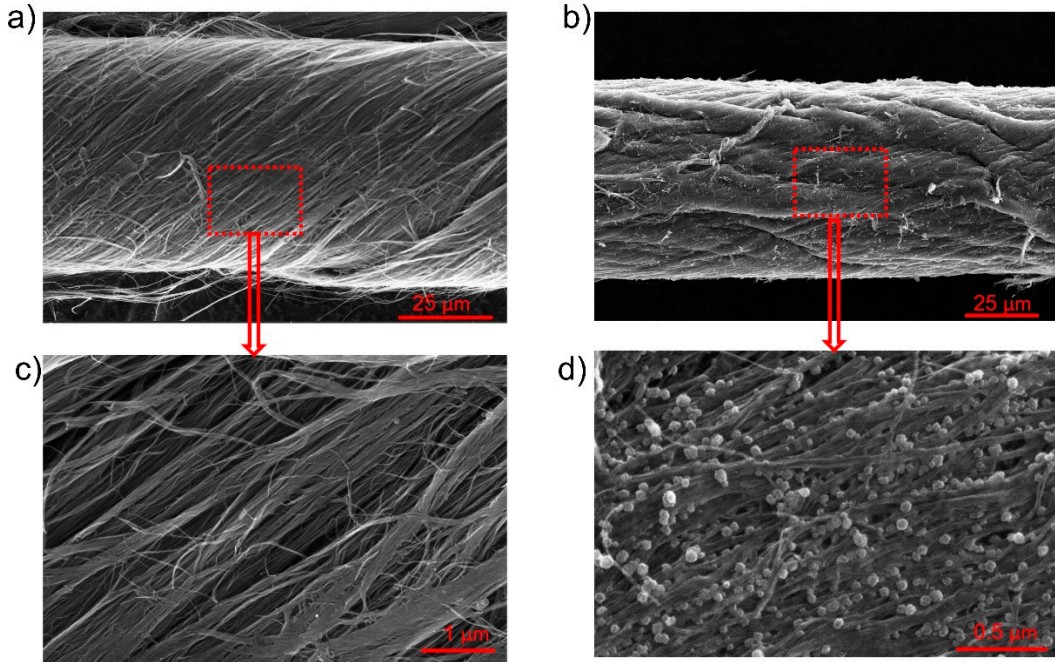

**Figure 2.** (**a**) SEM image of OPFCNT fiber; (**b**) SEM image of $FeC_2O_4$ + FeOOH coated OPFCNT fiber; (**c**) Higher magnification image of (**a**); (**d**) Higher magnification image of (**b**).

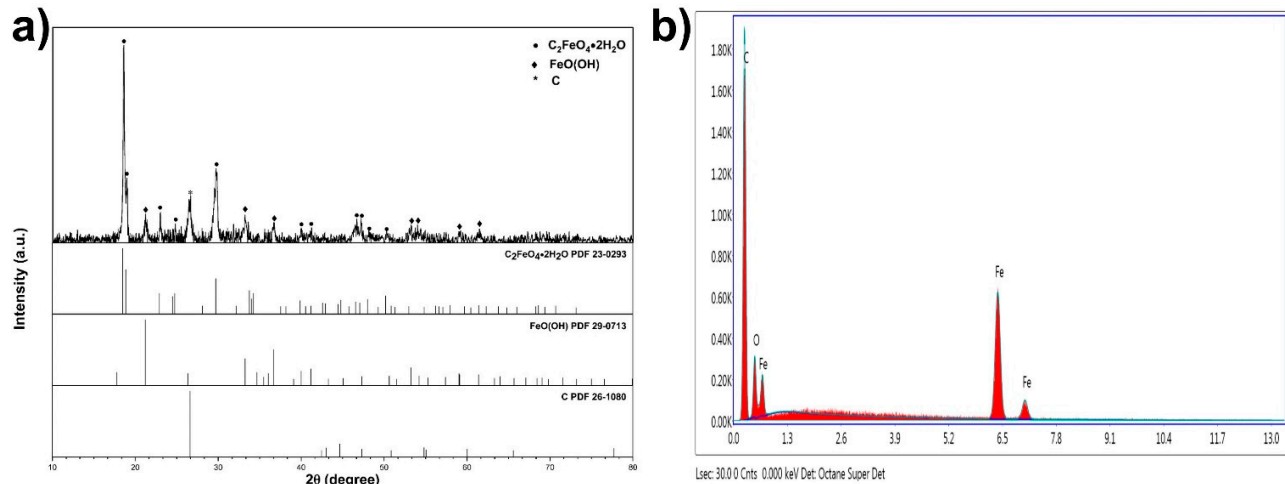

**Figure 3.** (**a**) XRD pattern of $FeC_2O_4$ + FeOOH OPFCNT fiber and (**b**) EDS of $FeC_2O_4$ + FeOOH fiber.

The Raman spectra of carbon-based materials, of which multi-wall CNTs are included, is characterized by two major peaks, D and G, at approximately 1360 cm$^{-1}$ and 1580 cm$^{-1}$, along with a 2G peak around 2700 cm$^{-1}$, respectively. The D band indicates defects created due to the formation of sp$^3$ bonds or single bonded carbons on the tube's surface. The G band, on the other hand, specifies the sp$^2$ hybridized carbon atoms with double bonds and represents the graphitic nature of the CNTs [51–53]. Figure 4a clearly shows the D and G peaks present in the Raman spectrum of the pristine CNT sample. However, in the $FeC_2O_4$/FeOOH CNT sample, we noticed a very high D peak and an almost non-existent G peak. We also observed Fe peaks within the range from about 200 cm$^{-1}$ to 750 cm$^{-1}$, which were not present in the pristine CNT fiber.

Further analysis of the composite CNT fibers has been conducted by XPS. The XPS data is presented in Figure 4b–d. From the obtained spectra we found the presence of carbon, oxygen, iron, and silicon peaks. The samples for XPS analysis were prepared by

wrapping the CNT fiber around a silicon wafer which explains the presence of the Si peaks. XPS peak deconvolution was conducted using a Shirley-type baseline and pure Gaussian line shape. The high-resolution Fe 2p spectrum in Figure 3c revealed peaks at 711.4 and 724.8 eV corresponding to Fe $2p_{3/2}$ and Fe $2p_{1/2}$, respectively. The O1s spectrum was also deconvoluted into three peaks at 534.2, 533.7, 530.78 eV, which were identified as C-OH, Fe-OH, and Fe-O-Fe, respectively (Figure 3d) [14].

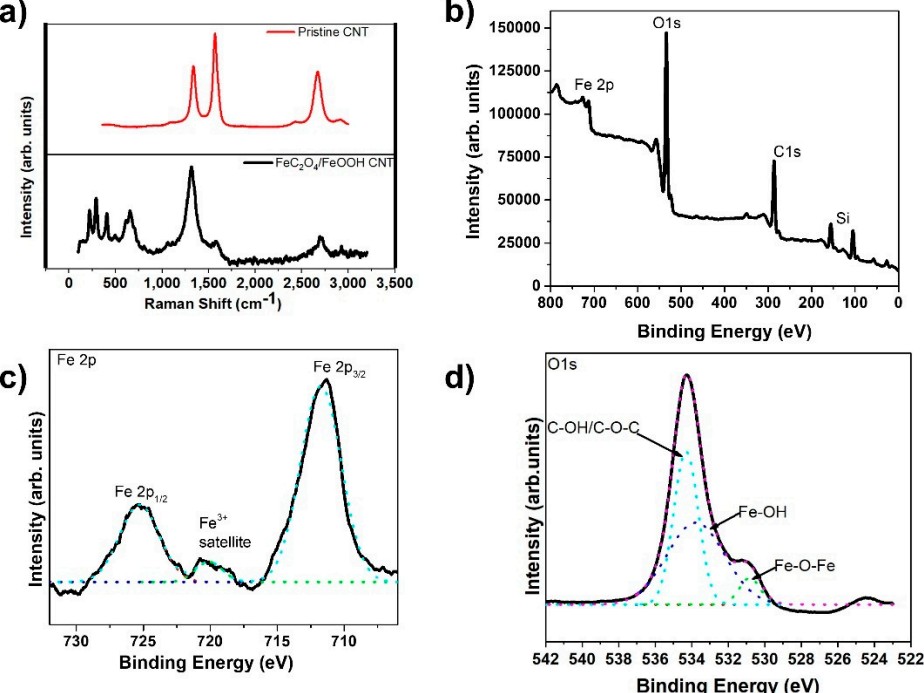

**Figure 4.** (**a**) Raman spectra of $FeC_2O_4$/FeOOH/CNT fiber; (**b**) XPS survey scan of $FeC_2O_4$/FeOOH/CNT fiber; (**c**) High Resolution spectra of Fe2p from (**b**); and (**d**) high resolution scan of O1s from (**b**).

Figure 5 presents the electrochemistry data for the anode material fabricated in this study. Four different anode materials were prepared and tested with mass loadings of $FeC_2O_4$/FeOOH: 0.026, 0.055, 0.094, and 0.174 mg/cm$^2$, respectively. These anode materials were then tested in a three-electrode arrangement using KOH as electrolyte, Ag/AgCl as the reference, and platinum wire as the counter electrode. Figure 5a displays specific capacitances (F/cm$^2$) vs. current density for the four types of anodes prepared with different mass loading. It is obvious from the graphs there that the 0.094 mg/cm$^2$ mass loaded device presents the best data in terms of the highest specific capacitance. The highest specific capacitance for this anode was 324 mF/cm$^2$ at 2 mA/cm$^2$. In Figure 5b we present a comparison of the gravimetric and areal capacitances of the fiber electrodes with various anode mass loadings at 2 A/g and 2 mA/cm$^2$ current densities, respectively. The data suggests that the optimized mass loading for this anode material is at 0.094 mg/cm$^2$. Finally, Figure 5c presents the cyclic voltammetry curves at various scan rates for the anode with mass loading of 0.094 mg/cm$^2$. A pair of oxidation and reduction peaks are observed, especially at the low scan rates attributed to the redox reactions taking place due to the presence of $FeC_2O_4$/FeOOH.

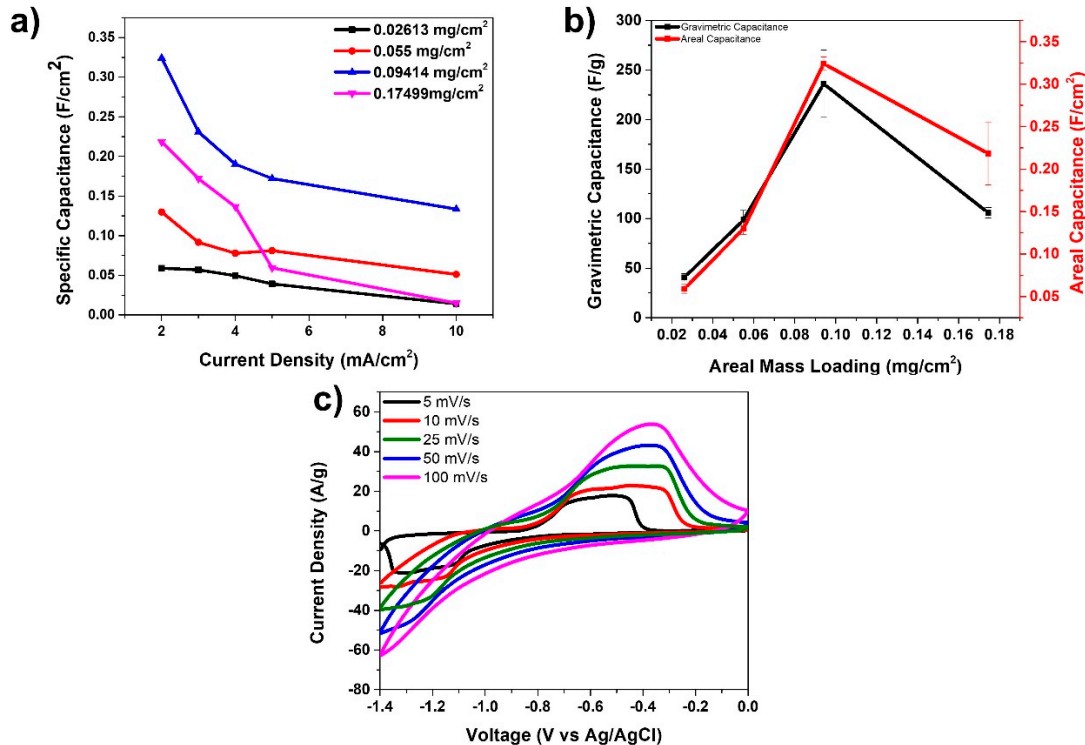

**Figure 5.** (**a**) Specific capacitance vs. current density for anode materials; (**b**) gravimetric capacitance and areal capacitance of various anodes at 2 A/g and 2 mA/cm$^2$; and (**c**) CV curves at various scan rates for 0.094 mg/cm$^2$ mass loading anode.

### 3.2. Positive Electrode (Cathode)

In Figure 6a,b we present SEM images of the PANI decorated CNT fibers. The formation of the PANI nanorods on the CNT fibers is seen under higher magnification in Figure 6b. There were two different cathode materials prepared by varying the oxidation polymerization time of PANI on the CNT fibers. We selected 15 and 60 min corresponding to mass loadings of 1.125 mg/cm$^2$ and 1.392 mg/cm$^2$, respectively. Figure 6c displays the Raman spectra of the CNT fiber with 1.392 mg/cm$^2$ PANI mass loading, which is compared with the Raman spectra of pure PANI, CNT fiber, and OPFCNT. We observed an increase in the D peak intensity at approximately 1360 cm$^{-1}$ for the OPFCNT compared to the CNT fiber, which is expected due to defects created during the oxygen plasma functionalization of the pristine material [47]. The same Figure 6c shows the presence of PANI peaks at ~1168 cm$^{-1}$, ~1349 cm$^{-1}$, and ~1498 cm$^{-1}$ on the CNT fiber with 1.392 mg/cm$^2$ PANI mass loading. These peaks correspond to the "in-plane" deformation of the C-C bond of the quinoid and the C-N and C=N stretching in the quinoid and benzenoid rings, as reported for PANI [54–56].

Figure 7 presents half-cell data obtained from the tested PANI-CNT cathodes. Figure 7a shows a CV curve of 60 min-PANI-CNT electrode tested at a scanning rate of 5 mV/s in 1 M H$_2$SO$_4$. It was observed that the pseudocapacitive material PANI on the surface of the CNT fiber handled a fast surface and near-surface redox [57–59]. Three oxidation and reduction peaks, labeled as O1/R1, O2/R2, and O3/R3, have been noticed on the CV curve and are consistent with reports in the literature. O1/R1 represents the redox transition of leucoemeraldine/emeraldine forms, O2/R2 denotes the hydroquinone/benzoquinone redox reaction, and O3/R3 represents the faradaic transformation of emeraldine/pernigraniline forms [60–62].

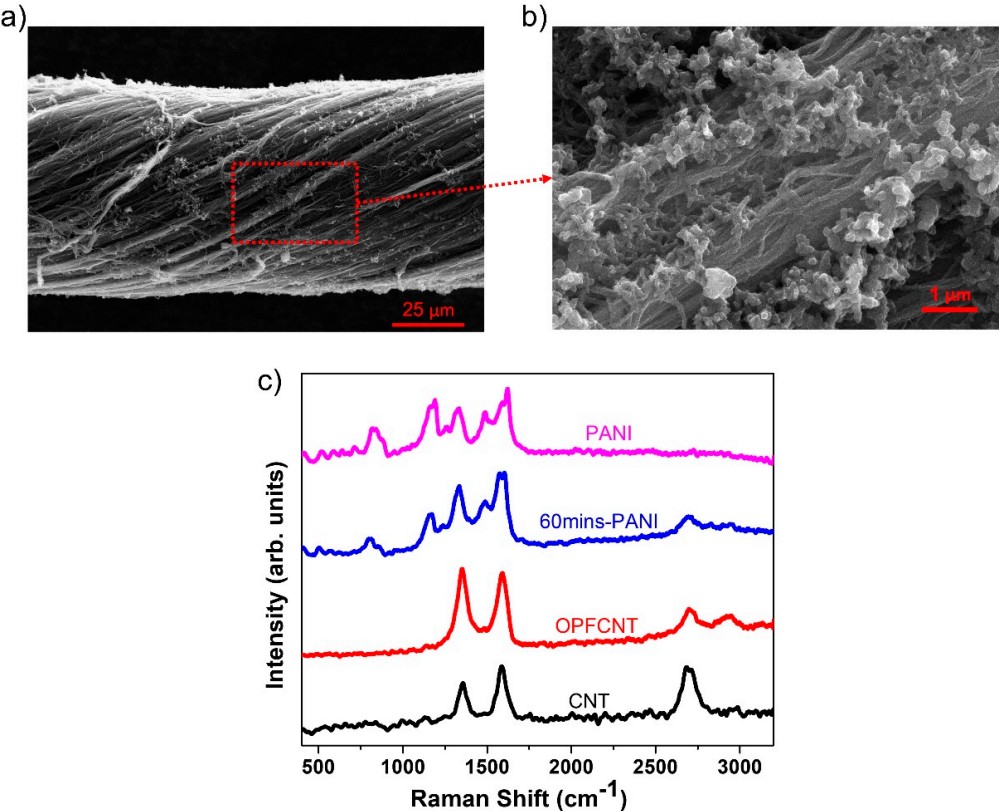

**Figure 6.** (**a**) SEM of 60 min PANI-CNT fiber; (**b**) enlarged image from (**a**); and (**c**) Raman of CNT, OPFCNT, 60 min PANI-CNT, and pure PANI.

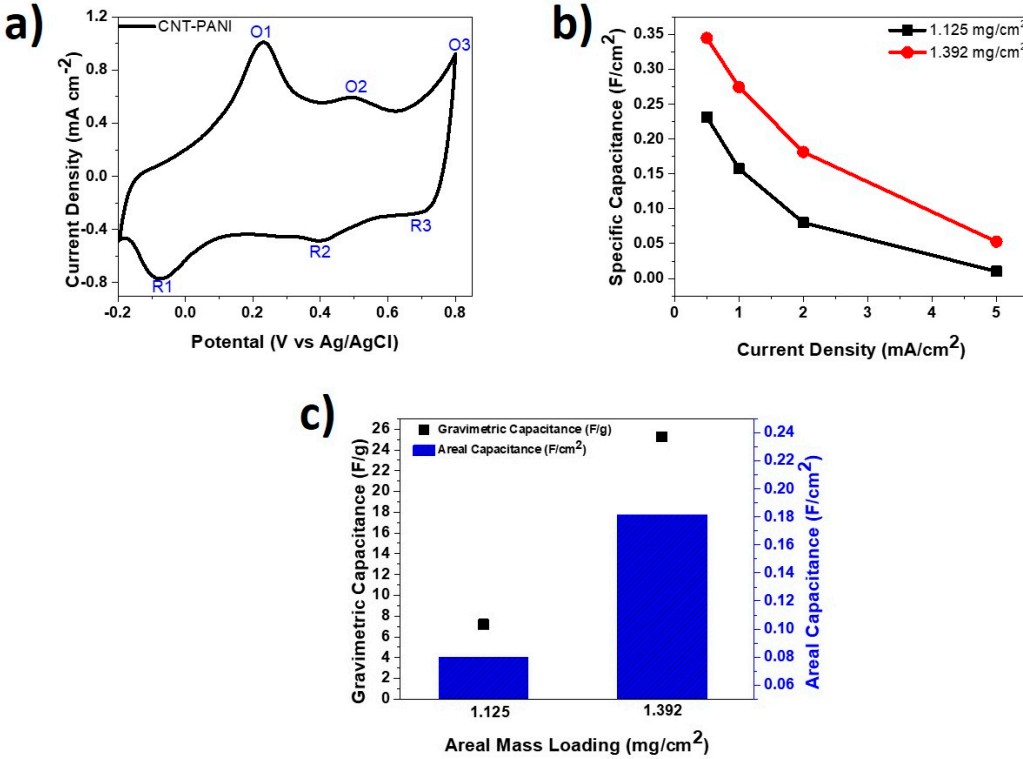

**Figure 7.** (**a**) CV curve of PANI-CNT fiber at 5 mV/s in $H_2SO_4$ electrolyte; (**b**) specific capacitance at various current densities for PANI-CNT fibers; and (**c**) areal and gravimetric specific capacitances for PANI-CNT fibers at current densities of 2 mA/cm$^2$ and 2 A/g, respectively.

Figure 7b compares the areal specific capacitances for the two PANI-CNT fibers with mass loadings of 1.125 mg/cm$^2$ and 1.392 mg/cm$^2$ at different current densities. The 1.392 mg/cm$^2$ PANI-CNT fiber attained specific capacitance of 344.9 mF/cm$^2$ at 0.5 mA/cm$^2$ compared to the 1.125 mg/cm$^2$ one, which showed 231.5 mF/cm$^2$ at the same current density. It was further observed that at 1.392 mg/cm$^2$ loading, there was consistently higher areal specific capacitance across all tested current densities. As displayed in Figure 7c, the 1.392 mg/cm$^2$ mass loaded PANI-CNT fiber achieved much greater gravimetric specific capacitance and areal specific capacitance compared to the 1.125 mg/cm$^2$ mass loaded fiber at current densities of 2 mA/cm$^2$ and 2 A/g, respectively.

## 4. Full-Device Characterization

A full, two-electrode cell was fabricated based on the CNT/FeC$_2$O$_4$ + FeOOH as an anode and the PANI-CNT fiber as a cathode. The CNT/FeC$_2$O$_4$ + FeOOH with mass loading of 0.094 mg/cm$^2$ and the PANI-CNT fiber with 1.392 mg/cm$^2$ mass loading were selected for this test due to their superior electrochemical performance, as seen in Figure 7b,c.

In order to create devices with good cyclic stability and columbic efficiency, and to maximize the available operating voltage window (OVW), we balanced the charge stored by the anode and the cathode, and calculated the mass ratio of the electrodes using the equations below:

$$Q = M\,C\,\Delta V \tag{1}$$

$$Q^+ = Q^-,\ M^+ C^+ \Delta V^+ = M^- C^- \Delta V^- \tag{2}$$

$$\frac{M^+}{M^-} = \frac{C^- \Delta V^-}{C^+ \Delta V^+} \tag{3}$$

where Q is the charge stored in the electrode, C is the gravimetric capacitance, $\Delta V$ represents the potential, and M is the total mass of the electrodes. The gravimetric capacitance was derived from three electrode testing. A mass ratio of $\frac{m^+}{m^-} = 0.085$ was calculated at 25 mV/s. Therefore, every 1 cm length of CNT/FeC$_2$O$_4$ + FeOOH fiber required matching with an approximately 12 cm long PANI-CNT fiber. The fabricated device was stable in PVA-KOH electrolyte up to 2.1 V and in PVDF-EMIMBF4 electrolyte up to 2.8 V.

In Figure 8a the cyclic voltammetry curves for the 1.23 M PVA-KOH device at various scan rates and after matching are shown. There are distinctive oxidation and reduction peaks, especially at lower scan rates due to the redox reactions taking place. Figure 8b displays a charge-discharge graph at a current density of 0.5 mA/cm$^2$ for the matched PVA-KOH device, which corresponds to a columbic efficiency of 86.49%. This device attained an energy density with values between 0.05 μWh/cm$^2$ and 4.07 μWh/cm$^2$ and a power density between 0.18 mW/cm$^2$ and 0.42 mW/cm$^2$. The obtained high cyclic retention in Figure 8c was 96.22% after 4000 cycles at a current density of 1 mA/cm$^2$ after the electrodes were matched. It was interesting to note that the lesser the concentration of the KOH, the greater the capacitance retention (Figure S3). Although the higher concentration KOH devices gave overall higher energy and power densities, the capacitance retention was greatly impacted.

Another device was fabricated with PVDF-EMIMBF4, and the obtained data is presented in Figure 9a–c. The CV curves at various scan rates are displayed in Figure 9a. These curves are quasi-rectangular, and the peaks observed previously due to redox reactions in the PVA-KOH devices are not apparent here. The charge discharge graph at a current density of 0.5 mA/cm$^2$ is shown in Figure 9b. The obtained columbic efficiency of 97.89% (Figure 9c) from this graph is much better than that for the PVA-KOH devices. However, the time of the whole scan in PVDF-EMIMBF4 electrolyte was approximately 7 s compared to the PVA-KOH device, which required about 25 s. Subsequently, the PVDF-EMIMBF4 device revealed more modest electrochemical data compared to the PVA-KOH one. The PVDF-EMIMBF4 device despite having a larger operating voltage window of 2.8 V, attained energy densities between 0.17 μWh/cm$^2$ and 3.06 μWh/cm$^2$, and power densities

between 0.26 mW/cm$^2$ and 0.97 mW/cm$^2$. As shown in Figure 9c, this device, however, demonstrated greater capacitance retention of 96.76% when tested over 4000 cycles at a current density of 1 mA/cm$^2$, compared to what was seen for the PVA-KOH device.

Any future application of the fabricated in this work energy storage devices is targeting wearable electronics. For such an application, testing the device performance on a frequently bending substrate such as fabric or polymer film is important. In Figure 10 data from bending tests of the created supercapacitor devices are presented. The related CV curves at a scanning rate of 200 mV/s for matched PVA-KOH and PVDF-EMIMBF4 devices undergoing multiple cycling bending up to 180 degrees are displayed in Figure 10a,b. The charge-discharge graphs at a current density of 0.5 mA/cm$^2$ for the same devices are shown in Figure 10c,d. It was obvious from all the graphs in Figure 10 that there was a complete overlap of the CVs and charge-discharge curves. This data illustrated the device bendability and flexibility over multiple bending cycles, which did not affect its performance. A comparison of these devices with others in literature is presented in Figure S2 (Ragone Plot) and Table S1 in the supplementary information.

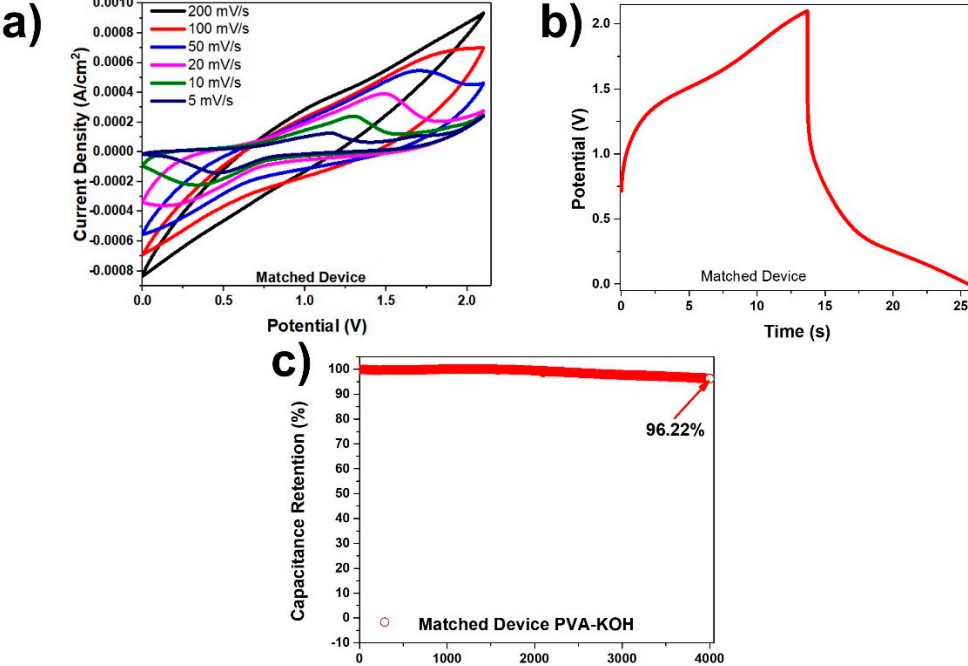

**Figure 8.** (**a**) CV curves at various scan rates for matched PVA-KOH device; (**b**) charge-discharge curve at a current density of 0.5 mA/cm$^2$ for the matched PVA-KOH device; and (**c**) capacitance retention of matched PVA-KOH device at a current density of 1 mA/cm$^2$ tested over 4000 cycles.

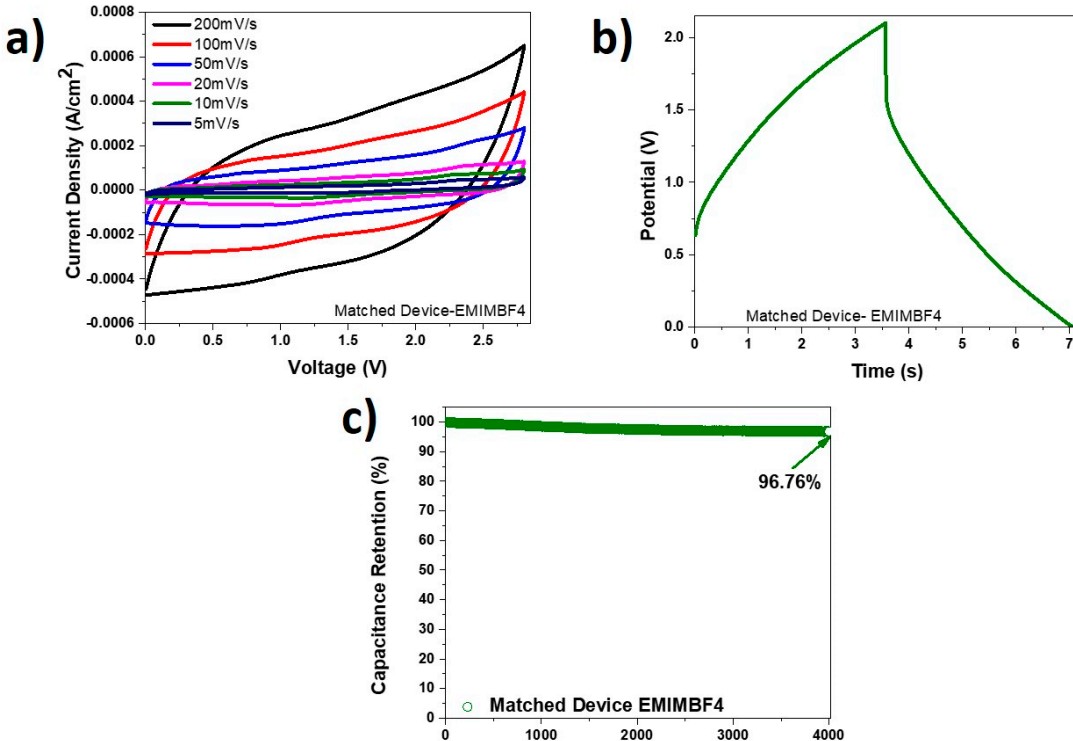

**Figure 9.** (**a**) CV curves at different scan rates for PVDF-EMIMBF4 device; (**b**) charge-discharge curve at current density of 0.5 mA/cm$^2$ for the PVDF-EMIMBF4 device; (**c**) capacitance retention of PVDF-EMIMBF4 device at current density of 1 mA/cm$^2$ tested over 4000 cycles.

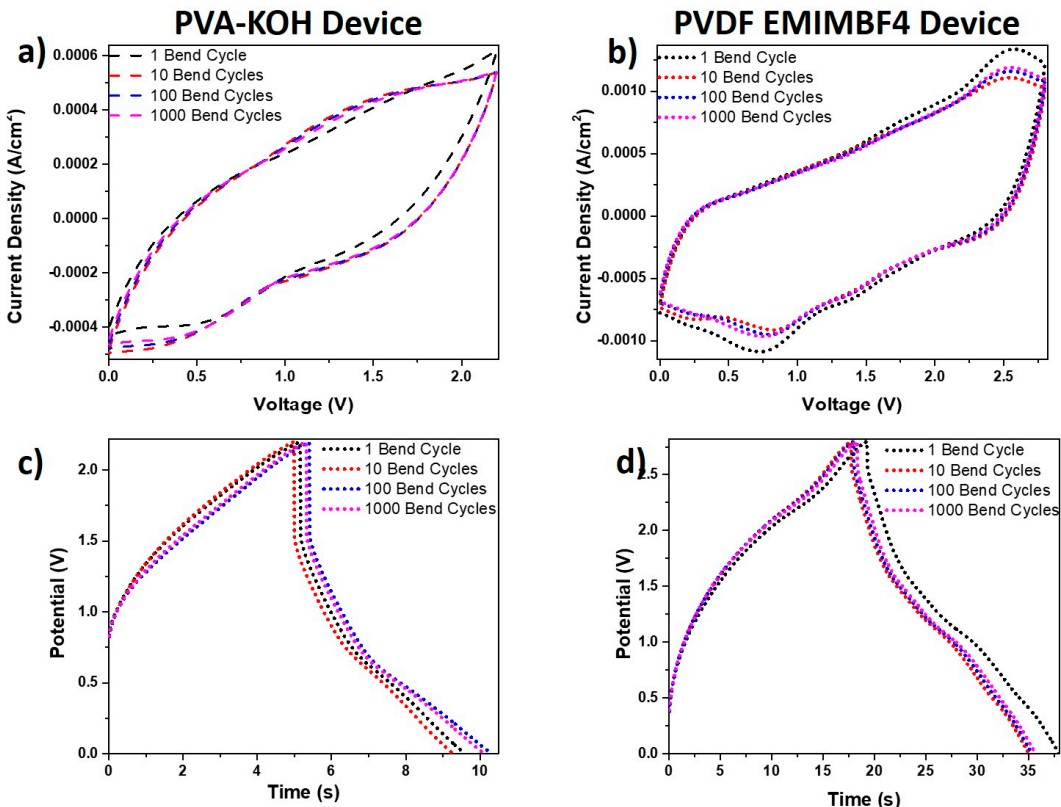

**Figure 10.** Cyclic voltammetry curves at a scanning rate of 200 mV/s obtained over multiple cycling bending up to 180 degrees for (**a**) the PVA-KOH device and (**b**) the PVDF-EMIMBF4 device. Charge-discharge graphs at a current density of 0.5 mA/cm$^2$ obtained over multiple bending cycles for (**c**) the PVA-KOH device and (**d**) the PVDF-EMIMBF4 device.

## 5. Conclusions

In this work, we have successfully designed, fabricated, and tested a high-energy density asymmetric fiber supercapacitor. It has been demonstrated that the anode can be made of nanoparticle-size $FeC_2O_4$ and FeOOH embedded on the surface of CNT fibers by a solvothermal process, whilst the cathode was a CNT-PANI composite fiber fabricated by oxidation polymerization.

These electrodes were characterized by Raman spectroscopy, XPS, XRD, SEM, and electrochemically. Two devices were fabricated using both PVA-KOH and PVDF-EMIMBF4 electrolytes. The device made from PVA-KOH attained a high energy density between $0.05 \ \mu Wh/cm^2$ and $4.07 \ \mu Wh/cm^2$ and a power density between $0.18 \ mW/cm^2$ and $0.42 \ mW/cm^2$. The device fabricated with PVDF-EMIMBF4 revealed energy densities between $0.17 \ \mu Wh/cm^2$ and $3.06 \ \mu Wh/cm^2$ and power densities between $0.26 \ mW/cm^2$ and $0.97 \ mW/cm^2$.

It has been found that devices made with PVA-KOH had poor cyclability, which was attributed to the redox reactions taking place. This conclusion was proven by testing devices made with different molarities of KOH. The performance of devices containing higher molarities was significantly influenced by the chemical reactions, resulting in more generated charges and consequently greater energy and power densities. This positive trend came at the expense of the poorer cyclability. The conducted study showed that there is a tradeoff between the device energy parameters and the capacitance retention. There is, therefore, a need to optimize the amount of KOH electrolyte used to attain high enough energy and power densities, as well as a good cyclability.

**Supplementary Materials:** The following are available online at: https://www.mdpi.com/article/10.3390/c7030062/s1. Figure S1: Schematic of the supercapacitor; Figure S2: Ragone plot comparing data generated by our device with data from other devices from the literature; Figure S3: (a) CV curve at a scanning rate of 5 mV/s highlighting the chemical reactions taking place during charge and discharge; (b) Capacitance retention data for KOH devices at different molarities of the electrolyte tested over 4000 cycles; Table S1: Comparison of energy density and power density data of our devices with others in literature.

**Author Contributions:** P.K.A., S.G. and V.S. conceived and designed the experiments; P.K.A., K.J., M.K., S.N.K., Y.-Y.H., G.Z., Y.F. and M.N. performed the experiments; P.K.A., S.N.K. and K.J. analyzed the data; P.K.A., S.N.K., Y.-Y.H. and V.S. discussed the data; P.K.A. and V.S. contributed reagents/materials/analysis tools. All authors have read and agreed to the published version of the manuscript.

**Funding:** This work was funded by a NASA Grant NNX13AF46A and partly by the National Institute for Occupational Safety and Health through the Pilot Research Project Training Program of the University of Cincinnati Education via the Research Center Grant # T42OH008432.

**Data Availability Statement:** Data is available upon request.

**Conflicts of Interest:** The Authors declare no conflict of interest.

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
