# Peer review of "Asymmetric Fiber Supercapacitors Based on a FeC2O4/FeOOH-CNT Hybrid Material"

_carbon, 2013_

Round 1

Reviewer 1 Report

This is an interesting manuscript dealing with the application of CNFs in asymmetric supercapacitors. Background and experimental approach, material characteristics have been described thoroughly. The manuscript could be published after revision:

  1. A comparison with established/commercial capacitor technologies is missing

  1. A photo or sketch of the device design would be helpful to the readers.

  1. In Figure 7a, the current decreases with increasing scan rate which does not make sense.

Author Response

Attached are answers to the reviewers comments

Reviewer 2 Report

This manuscript entitled “Asymmetric Fiber Supercapacitors Based on a FeC2O4/FeOOH-CNT Hybrid Material” [C] Manuscript ID: carbon-1292857, by Adusei et al. report synthesis of Carbon Nanotube (CNT) and iron oxalate + iron hydroxide based composite yarn, via solvothermal processes that attained energy densities in the range of 0.17 μWh/cm2 and 3.06 μWh/cm2, and power densities between 0.26 mW/cm2 and 0.97 mW/cm2, when assembled in a supercapacitor with PVDF-EMIMBF4 electrolyte. Such devices can be used for powering futuristic wearable electronics. This work is publishable on [C] after a major revision by fully considering the following suggestions:

  1. Novelty and purpose of this work is not clear. There are several reports on CNT yarn-based supercapacitors which have already reported far superior performance than the present work. For instance: Fenghua Su et al. 2014 (doi.org/10.1002/smll.201401862) reported CNT@Co3O4 composite yarns display excellent electrochemical properties with very high capacitance of 52.6 mF cm−2 and energy density of 1.10 μWh cm−2. While Changsoon Choi et al 2018 (doi: 10.1039/C8RA01384E) reported MnO2-CNT based yarn which exhibited wider potential window (up to 3.5 V) and resulting high energy density (43 μW h cm−2). Authors should explain the importance of such low performance device if high performance and more stable devices are already in literature.
  2. High energy density SC presents poor retention of 18.57%. Such device can not present any advantage over other reports. Then how this created unique compo-sition of iron oxalate + iron hydroxide + CNT as an anode is advantageous?
  3. Authors should compare their work with the literature in a tabular form.
  4. What is the advantage of using iron oxalate + iron hydroxide, if more economical MnO2, NiO or Co3O4 etc. can achieve higher energy storage ?
  5. Section 2.1. Fabrication of CNT / OPFCNT Fiber should be “Fabrication of CNT and OPFCNT Fiber.
  6. Section 2.2. FeC2O4 + FeOOH Deposition on CNT Fiber for Anode Formation. Here, what kind of CNT fiber is used: CNT or OPFCNT Fiber. Kindly mention this. Also authors have not used any special oxidizing agent with Fe(NO3)3.9H2O, than how FeC2O4 + FeOOH are formed ? Write mechanism of this complete process.
  7. Section 2.3. PANI Deposition on CNT Fiber for Cathode Formation. “1 M aniline monomer dissolved in 1 mol/L HCl and 1 M ammonium perox-ydisulphate (APS) in 1 mol/L HCl, as reported in our previous work [32]. The hybrid fiber was created at a 2:1 ratio (aniline monomer to APS) over various periods of time from 15 to 60 minutes”. This line is confusing.
  8. Figure 2e and f can be presented in same tile size as the a-d. Scale should be redrawn and magnified.
  9. “Looking at high magnification SEM images (Figure 2d), we were unable to see pronounced individual carbon nanotubes or nanotube bundles”. This statement is not correct. Bundles of CNT are clearly visible in Figure 2d.
  10. “The Raman spectra of multi-wall CNTs is characterized by two major peaks: D and G”, this statement is true for any CNT, not only multi-walled. Kindly correct.
  11. Figure 3 a both Raman spectrums should be clubbed in one figure. The symmetry of Figure 3 is looking poor.
  12. Figure 4b and Blue curve of Figure 4a are presenting same information. Why to present same figure again?
  13. Figure 4 d) CV curves at various scan rates for 0.094mg/cm2 mass loading anode. Literature suggest, usually aqueous electrolyte has a potential window of ± 1.2 V. How the CV shows stability till -1.4 V. Explain this mechanism.
  14. “The formation of the PANI nanorods on the CNT fibers is seen under higher magnification in Figure 5b” This statement is not true. No nanorod structure can be observed from Figure 5b. Kindly correct it.
  15. What is the loading percentage for Pani over CNT ? Pani is only deposited on the surface of CNT not interstitial site; will it be advantageous? How?
  16. Figure 7 b clearly expose the IR drop during Charge -discharge process. IR drop here is approx. 0.7 V, which is very high. How a cell with such large IR drop can be advantageous? Similar is Figure 8b.
  17. A columbic efficiency of 86.49% is not very good. CNT based SCs show much better columbic efficiency. Explain.
  18. Typos: Vesselin Shanov a and b *, * Correspondence: Corresponding, PVDF-EMIMBF4 electrolyte The, CNT fibers; FeC2O4/FeOOH-CNT etc..
  19. Authors should cite latest literature. Most of the reference presented are before 2015.

Author Response

Attached are the answers to the comments
